# Global Distribution Patterns and Climatic Drivers of Plant Diversity in Rubiaceae

**DOI:** 10.3390/biology14121719

**Published:** 2025-12-01

**Authors:** Sihong Yang, Yangyang Liu, Yuvenalis Morara Mbuni, Weirong Huang, Yadong Zhou, Huan Zhang

**Affiliations:** 1School of Life Sciences, Nanchang University, Nanchang 330031, China; 2East African Herbarium, National Museums of Kenya, Nairobi P.O. Box 45166-00100, Kenya

**Keywords:** Rubiaceae, diversity, species richness, growth form, environment factors, actual evapotranspiration

## Abstract

As the fourth largest family of angiosperms, Rubiaceae is a key component of tropical and subtropical plant taxa. However, in-depth research on its geographical distribution patterns and potential environmental driving mechanisms at the global scale is still lacking. In this study, we integrated Rubiaceae species distribution data with relevant environmental data, taking the entire Rubiaceae family and its different growth forms as research objects to explore their geographical distribution patterns and underlying environmental driving mechanisms. This work provides theoretical support for subsequent investigations into the evolution, ecological adaptation, and conservation of Rubiaceae.

## 1. Introduction

Exploring the complex patterns and underlying drivers of biodiversity is the core theme of macroecology and biogeography [1,2]. The geographical patterns of plant diversity have captivated the attention of ecologists and biogeographers for over two centuries. Classic theories, such as “the tropics are the cradle of biodiversity” exemplified by the contributions of Humboldt [3] and Wallace [4], reveal the existence of a prominent Latitudinal Diversity Gradient (LDG) in global plant diversity. This gradient is characterized by a decline in plant diversity with increasing latitude, peaking in the stable, warm, and humid climates of the tropics. Species richness, a key biodiversity metric widely used in numerous research [5,6], offers critical insights into species coexistence, resource allocation, and environmental adaptation [7]. Unraveling global-scale geographic gradients in species richness and their environmental drivers has become increasingly urgent under anthropogenic climate change, providing important insights for biodiversity conservation [8,9,10,11].

Current climate, historical climate dynamics, and human activities are widely recognized as significant environmental variables influencing variations in species richness patterns [12,13,14,15,16,17]. Among these factors, climate, including precipitation and temperature, are key drivers explaining global terrestrial species richness patterns [18]. Temperature and precipitation significantly influence species richness patterns across different plant groups at regional and global scales. For instance, liana diversity in tropical regions demonstrates that dry-season length (DSL) and precipitation influence tropical forest species richness through direct physiological constraints, particularly drought stress [19]. In addition, several studies have shown that current patterns of species richness are to some extent influenced by historical climate changes, because they may affect species formation, extinction, and dispersal [20,21,22,23]. During the Late Quaternary glacial–interglacial cycles, regions with more stable climates often serve as refugia, characterized by lower species extinction rates and higher speciation rates compared to regions with unstable climates [23,24,25]. Moreover, human activities exert multifaceted impacts on the composition, structure, function, and evolutionary processes of species diversity [26,27,28]. For instance, human activities have significantly shaped the unique biogeographic patterns of lizard communities across Caribbean islands by driving inter-island introductions of non-native chameleons, weakening the natural filtering effect of geographic isolation, and reshaping dispersal pathways through altered economic connectivity [29]. Therefore, a comparative analysis of how climate and anthropogenic disturbance influence phytogeographic distribution patterns may provide novel insights into the mechanisms underlying the maintenance and conservation of plant diversity.

Plants with different growth forms exhibit variation in morphological and life cycle, resulting in differential adaptive capacities in response to environmental changes. Herbaceous and woody plants are two fundamental categories representing these distinct growth forms [30]. Compared to woody plants, herbaceous plants usually have a relatively short life cycle, thereby accumulate genetic changes quickly, and possess strong adaptability to the environment and the ability to rapidly settle in various ecological niches [31,32,33]. In contrast, herbaceous plants have shorter generation times than woody plants, and in relatively recent evolutionary history, the flora evolution of herbaceous plants has experienced accelerated species formation compared to that of woody plants [34,35,36]. Until now, most research has focused on the geographic patterns of woody plant richness [37,38,39], yet herbaceous plants are often ignored.

Rubiaceae, a core eudicot family, constitutes the sister group to the remaining families of Gentianales [40,41,42]. It ranks as the fourth-largest flowering plant family, comprising 580 genera and over 14,000 species [43]. Rubiaceae are widespread and occur in all major regions of the world except the Antarctic Continent, but are predominantly a group in the tropics with greatest diversity in low- to mid-altitude humid forests [44]. This family exhibits diverse growth forms, including herbaceous plants, shrubs, trees, and lianas, with certain lineages possessing specialized ecological adaptations [45], making this family an ideal group for investigating distribution patterns of different growth forms and their underlying environmental drivers. At present, significant progress has been made in the research of plant diversity in the Rubiaceae family in some regions, such as Cameroon in Africa [46] and the study of phylogenetic relationships focusing on certain taxa [47,48]. However, there are still shortcomings in the study of species diversity and geographical distribution patterns at the global scale, which limits scholars’ understanding of the global diversity pattern of Rubiaceae species.

In this study, we utilized species distribution data of Rubiaceae species across 581 global geographic units and taxonomic data of different growth forms (herbaceous and woody plants) to assess global species richness patterns for both the entire family and its distinct growth forms. Our objectives are to (i) delineate the geographic distribution patterns of Rubiaceae species diversity at the global scale; (ii) investigate the distribution patterns of species diversity across different growth forms within Rubiaceae; (iii) analyze how four types of environmental variables (water and energy, climatic seasonality, historical climate change, and human influence) affect the geographic diversity patterns of distinct Rubiaceae growth forms, and explore their different driving mechanisms. This research not only provides a theoretical foundation for the conservation and sustainable utilization of Rubiaceae species, but also offers novel perspectives for enhancing the understanding of plant diversity distribution patterns and their underlying drivers.

## 2. Materials and Methods

### 2.1. Species Distribution Data

Our study based on 581 geographical units, most of which represent Level 4 units (“Basic Recording Units”) as defined by the International Working Group on Taxonomic Databases (TDWG; http://www.tdwg.org/ accessed on 15 January 2025) [49]. Species distribution data were sourced from the World Checklist of Vascular Plants (WCVP; https://powo.science.kew.org/ accessed on 15 January 2025), the Global Biodiversity Information Facility [50], and World Plants: Plant List (WP; https://www.worldplants.de/ accessed on 16 January 2025). Species names were standardized using the U.Taxonstand software package according to WCVP [51]. As a result, we compiled the distribution information for 14,630 Rubiaceae species.

Rubiaceae species were further divided into herbaceous and woody plants based on several databases such as the TRY Plant Trait Database (TRY; https://www.try-db.org/ accessed on 22 October 2024), World Checklist of Vascular Plant (WCVP; https://powo.science.kew.org/ accessed on 22 October 2024) and the Botanical Information and Ecology Network (BIEN; http://bien.nceas.ucsb.edu/bien/ accessed on 22 October 2024), floras (e.g., *Flora of China* and *Flora of North America*), and relevant studies [32,52,53,54]. Woody species are defined as those with a prominent aboveground stem that persists under changing environmental conditions [32], including trees, shrubs, and lianas. Herbaceous species include annual herbs, perennial herbs, and herbaceous vines. This definition has been used in multiple studies on plant diversity classification of different growth forms [55,56,57]. Among all species, nearly 80% are categorized as woody plants, and the rest as herbaceous plants.

### 2.2. Environmental Data

To comprehensively explore the impacts of environmental factors on the species diversity of Rubiaceae, we selected three categories of environmental data for analysis: current climate, historical climate change, and human influence. Current climate data include annual mean temperature (AMT), minimum temperature of coldest month (MTCM), annual precipitation (AP), precipitation of driest month (PDM), temperature seasonality (TS), precipitation seasonality (PS), and actual evapotranspiration (AET). AET data were obtained from CGIAR-CSI (https://cgiarcsi.community/data/global-high-resolution-soil-water-balance/ accessed on 25 October 2024), and other current climate variables were derived from the WorldClim database (https://www.worldclim.org accessed on 25 October 2024) at a resolution of 30 arc-seconds [58]. Historical climate change data include temperature change (AMT anomaly, AMTano) and precipitation change (AP anomaly, APano). AMTano and APano represent the differences in annual mean temperature and annual precipitation between the Last Glacial Maximum and the present, respectively. The temperature and precipitation data for the Last Glacial Maximum (LGM) were reconstructed using the MPI-ESM-P model and retrieved from the WorldClim database (http://www.worldclim.org, version 1.4 accessed on 25 October 2024) at a spatial resolution of 2.5 arc minutes. Human influence data include the Global Human Influence Index (HII), which was obtained from the Last of the Wild database (https://ciesin.columbia.edu/content/data accessed on 25 October 2024) for the period from 1995 to 2004. Details of species distribution data and environmental data can be found in Appendix A.

### 2.3. Data Analysis

The study used ArcGIS 10.8 to map the patterns of species richness (SR) and species density (SD) for Rubiaceae. Specifically, we generated global maps of total Rubiaceae species richness (SR_total_), the proportion of Rubiaceae species richness to the total vascular plant species richness (Proportion), herbaceous Rubiaceae species richness (SR_herb_), woody Rubiaceae species richness (SR_woody_), and the ratio of herbaceous Rubiaceae species richness to total Rubiaceae species richness (Ratio). Since species richness is typically linearly related to the log-transformed sampling area [59], we accounted for area effects by dividing the number of species in each region by the log10-transformed area (in square kilometers, with original areas uniformly expanded by tenfold for excessively small geographic units). This yielded global maps of total Rubiaceae species density (SD_total_), herbaceous Rubiaceae species density (SD_herb_), and woody Rubiaceae species density (SD_woody_).

We employed Ordinary Least Squares (OLS) regression to quantitatively assess the influence of environmental variables on global patterns of species richness and species density in Rubiaceae. Due to the right skewed distribution of SR, SD, TS, and PS, logarithmic transformation is applied to them. To reduce collinearity among predictor variables, we first divided all environmental variables into four groups: water and energy group (AMT, MTCM, AP, PDM, AET); climate seasonality group (TS and PS), and historical climate change group (AMTano and APano); human influence group (HII). Based on the calculation of Pearson’s correlation coefficients between each group of environmental factors, we retained only one variable from pairs with a correlation coefficient |r| > 0.8. Then, we conducted OLS analysis and spatial simultaneous autoregressive (SAR) analysis on the retained environmental factors to determine their relative effects on SR_total_, SR_herb_, SR_woody_, and Ratio. For SD_total_, SD_herb_, and SD_woody_, we performed the same analysis. After testing, the variance inflation factors (VIFs) of all variables included in the models were less than 4, indicating that the multicollinearity among the environmental variables has been resolved. Relevant information can be found in the Appendix A. Finally, we explored the independent and collective effects of four groups of environmental variables on the species richness and density of Rubiaceae plants using variance partitioning analyses. The above calculation process was carried out in R 4.2.1.

## 3. Results

### 3.1. Global Geographic Patterns of Diversity in Rubiaceae

We found that the SR and SD of Rubiaceae plants exhibit significant spatial heterogeneity on a global scale. SR_total_ peaked in tropical regions such as the Amazon Basin, Congo Basin, and Southeast Asia (Figure 1a), while declining toward boreal zones including Northern Europe, Siberian Russia, and Northern Canada (Figure 1a). Similarly, Ratio was highest in tropical areas like the Amazon Basin, Congo Basin, and Southeast Asia (Figure 1b), but significantly lower in boreal regions such as Northern Europe, Siberian Russia, and Northern Canada (Figure 1b). Additionally, SD_total_ peaked in tropical latitudes across South America, Central Africa, and Southeast Asia (Appendix A), whereas mid-to-high latitude regions exhibited reduced density, particularly in northern North America, Northern Europe, and Northern Asia (Appendix A).

### 3.2. Geographic Patterns of Diversity in Rubiaceae Across Different Growth Forms

Our study found significant differences in patterns of SR and SD among different growth forms in Rubiaceae. Herbaceous plants had a widespread global distribution. High values of SR_herb_ and SD_herb_ were primarily concentrated in tropical and subtropical regions such as South America, central Africa, and Southeast Asia, whereas low values were predominant in high-latitude areas like northern Europe, Siberian Russia, and northern Canada (Figure 2a; Appendix A). By contrast, woody plants exhibited a more concentrated distribution. High values of SR_woody_ and SD_woody_ were mainly in tropical and subtropical regions such as South America, central Africa, and Southeast Asia, and in mid-to-high-latitude regions such as Europe, North America, and central-northern Asia; however, these values were significantly lower, and their occurrence was much sparser (Figure 2b; Appendix A). Notably, the Ratio was relatively high in mid to high latitude regions such as North America and Europe, while it was relatively low in South America, central Africa, and other regions (Figure 2c).

### 3.3. Correlation Analysis of Environmental Factors

We calculated the standardized regression coefficients of SR and SD based on the OLS model and SAR model., and analyzed the effects of seven environmental single predictive factors on them. Accounting for spatial autocorrelation, the analysis results indicate that these seven individual environmental predictors collectively explain 39.1% of the variation in SR_total_ (Figure 3a) and 48.0% of the variation in SD_total_ (Appendix A). Among these predictors, AMT, AET, PS_log and HII have highly significant effects on the variation in both the SR_total_ and SD_total_ (Figure 3a; Appendix A). Specifically, AET shows a significantly positive correlation (*p* < 0.001), while HII shows a significantly negative correlation (*p* < 0.01). Among different growth forms (herbaceous, woody), the explanatory power of seven environmental single predictive factors for the variation in SR_herb_ is 18.8% (Figure 3b), and the explanatory power for the variation in SD_herb_ is 18.7% (Appendix A). AET, TS-log, PS_log and HII have significant effects on the variation in SR_herb_ and SD_herb_. The explanatory power on the variation in SR_woody_ is 60.6% (Figure 3c), and the explanatory power on the variation in SD_woody_ is 63.1% (Appendix A). Among them, AMT, AET, PS-log, and HII have significant effects on SR_woody_ and SD_woody_ (Figure 3c; Appendix A). However, in SD_woody_, AET shows a positive correlation and extremely significant (Appendix A). The explanatory power of seven environmental factors on the variation in Ratio both are 74.8% (Figure 3d; Appendix A). Different environmental factors have varying degrees of influence on the variation in the models, and all of them have significant effects on model variation. In addition, the effects of various environmental variables on SR and SD identified through OLS analysis were consistent with those derived from SAR analysis (Appendix A; Appendix A).

### 3.4. Different Environmental Variables Explain the Variation in Diversity in Rubiaceae

The joint explanatory power of four different environmental variables (water and energy, climate seasonality, historical climate change, and human influence index) for the variation in SR_total_ is 32.1% (Figure 4a), and the joint explanatory power for SD_total_ is 41.7% (Appendix A). Among them, the independent explanatory power of water and energy for the variation in the SR_total_ and SD_total_ is 16.6% and 16.5%, respectively, which is the strongest. In different growth forms (herbaceous, woody), there are differences in the joint and independent explanatory power of four different environmental variables for their respective SR and SD. In the SR_herb_ and SD_herb_, the four environmental variables can jointly explain 9.8% and 9.1%, respectively (Figure 4b; Appendix A). The independent explanatory power of water and energy for the variation in the SR_herb_ and SD_herb_ is 6.7% and 7.5%, respectively. However, the independent explanatory power of the historical climate change for SR_herb_ is almost unexplainable, ranking the lowest among all predictors. In contrast, historical climate change and human influence index exhibit the weakest independent explanatory power for SD_herb_ variation. For woody species, the four environmental variable groups collectively explain 58.6% of the variation in the SR_woody_ (Figure 4c) and 61.0% of the variation in the SD_woody_ (Appendix A). Water and energy demonstrated the strongest independent explanatory power for SR_woody_ and SD_woody_ variation, whereas the historical climate change exhibits the weakest independent explanatory power for them. In addition, the combined explanatory power of the four environmental variable groups is both 75.1% for their Ratio variation (Figure 4d; Appendix A). Among these groups, water and energy contributes the strongest independent explanatory power to the model variation, while the historical climate change exhibits the lowest independent explanatory power.

## 4. Discussion

### 4.1. Global Geographic Pattern of Rubiaceae Diversity

Understanding the spatial distribution of angiosperms is essential to designating meaningful diversity centers, preserving plant resources, and maintaining ecosystem functions [60,61,62]. The geographical pattern of overall Rubiaceae species diversity in this study reveals that the majority of Rubiaceae species are concentrated in the pantropical regions. This indicates that Rubiaceae species are exceptionally important for all types of tropical vegetation, particularly within tropical rainforests [63]. These findings align with previous studies, which identified tropical and subtropical regions as the core distribution areas for Rubiaceae species, with a marked decline in diversity towards temperate and polar zones [44,64].

The SR and SD for different growth forms (herbaceous and woody plants) exhibit divergent geographical patterns at the global scale. Notably, Rubiaceae herbaceous species exhibit a wider distribution range than their woody counterparts. In mid- to high-latitude regions, such as Europe, North America, and north-central Asia, herbaceous representatives of Rubiaceae are documented, whereas woody species are scarcely observed. In addition, the pattern of Ratio shows that herbaceous plants have high species richness in mid to high latitude regions, which is consistent with their relatively short life cycle, strong adaptability to the environment, and ability to quickly adapt to various ecological niches [31,32,33]. Our study reveals that woody plants are widely distributed across tropical and subtropical regions. This finding is consistent with prior observations that Rubiaceae woody plants are a major component of the tropical rainforest understory at low to mid-elevations [44]. The observed divergence in diversity patterns between growth forms can be attributed to their fundamentally distinct adaptive strategies to environmental variation. For instance, Smith and Donoghue [35] observed that herbaceous and woody plants exhibit differential genetic adaptive potentials in response to environmental changes due to variations in generation times and molecular evolutionary rates.

### 4.2. Differential Impacts of Environmental Factors in Rubiaceae

Although contemporary climate influences diversity patterns of various taxonomic groups [65], our findings demonstrate that current climate, historical climate change, and human influence index collectively account for significant contributions to the geographical distribution patterns of Rubiaceae diversity, with AET emerging as the strongest single predictor. This finding aligns with established research showing that AET explains vascular plant species richness patterns at both global and continental scales [10,66]. The difference is that in the SR_herb_ and SD_herb_, AET is the strongest single predictor, followed by TS-log. In the SR_woody_ and SD_woody_, AET is also the strongest single predictor; however, AMT and PS-log are other factors that exert a significant impact. Furthermore, AET exerts a stronger influence on Rubiaceae woody plants than on herbaceous plants, a pattern likely attributable to the superior adaptive capacity of herbaceous plants in resource-limited environments characterized by cold temperatures, aridity, and nutrient deficiency [32,67].

Beyond AET, the divergent influences of PS-log and AMT on their respective SR and SD models could be ascribed to the distinct ecological niches inhabited by plants of varying growth forms. Such differences may lead to contrasting adaptive strategies in response to environmental factors. Compared with herbaceous plants, woody plants tend to be relatively tall, have wider conductive vessels, and are generally more vulnerable to coldness and/or drought due to stronger sensitivity to water transport limitations, such as embolism [68,69,70]. Concurrently, seven environmental factors differentially influence variations in herbaceous Rubiaceae ratio. Notably, the seven selected environmental variables exhibited generally low explanatory power for SR_herb_ and SD_herb_. This indicates that the unique geographical distribution pattern of Rubiaceae herbaceous cannot be adequately explained by these broad environmental variables, implying the existence of potential unconsidered environmental factors shaping their distribution dynamics. This finding aligns with the conclusions of Ullah et al. and Das et al., who emphasized that soil properties, topographic features, elevation, and biotic interactions play dominant roles in regulating the geographical distribution patterns of herbaceous plants [71,72].

### 4.3. The Impact of Different Environmental Variables in Rubiaceae

In the current global context at regional scales, water and energy, climate seasonality, historical climate change, and the human influence index play crucial roles in shaping the differentiated geographical patterns of angiosperm diversity. Water and energy emerge as the dominant factors in model variation, followed by climate seasonality. The water-energy dynamics hypothesis [13] has been shown to explain a wide range of phenomena [39,73]. Our research also reveals that water-energy dynamics emerge as dominant drivers of diversity distribution patterns across different growth forms of Rubiaceae species [55,74]. Interestingly, in the geographical patterns of Rubiaceae diversity across different growth forms, we found that woody plant diversity exhibited stronger responses to LGM (Last Glacial Maximum), climate change than herbaceous plant diversity. This divergence may reflect the shorter generation times and weaker climatic niche conservatism in herbaceous plants [31], which enables them to adapt to novel environments more effectively than woody plants. Furthermore, in our study, climatic seasonality emerged as the second most influential single variable and a key underlying driver determining Rubiaceae diversity patterns. This may be linked to the niche conservatism of Rubiaceae plants adapted to tropical climates. Although our study identified the key environmental drivers shaping the distribution patterns of Rubiaceae species, it is critical to acknowledge the potential limitations of the GBIF dataset. Notably, the selected environmental factors exhibit broad spatial consistency across different geographical units. Thus, these sampling biases are unlikely to substantially alter our core conclusion that water and energy, and climatic seasonality collectively exert dominant control of Rubiaceae distribution.

## 5. Conclusions

This study investigated the diversity of the Rubiaceae species at the global scale, as well as the geographical distribution patterns and driving factors of different growth forms. Our results show that Rubiaceae species are widely distributed in tropical and subtropical regions, with relatively few in temperate and polar regions, exhibiting a clear distribution pattern of latitude gradient diversity. Furthermore, herbaceous and woody plants exhibit markedly divergent patterns in species richness and species density, reflecting superior adaptive strategies of herbaceous plants in rapidly changing environments compared to woody plants. Multiple environmental variables significantly influence the distribution patterns of Rubiaceae as well as specific growth forms. Among these factors, actual evapotranspiration (AET) emerges as the strongest single predictor explaining diversity pattern variations. Concurrently, water and energy and climatic seasonality appear to be primary drivers underlying the geographical distribution differences in overall Rubiaceae, herbaceous and woody plant diversity. Consequently, based on these research findings, we can better protect Rubiaceae species and preserve the existing biodiversity patterns. However, our access to environmental variables influencing the distribution patterns of Rubiaceae species remains limited, leaving certain facets of their current distribution unexplained. In particular, the unique distribution patterns of herbaceous species remain poorly understood. In addition, future research should further investigate these unresolved aspects and elucidate the distinct distribution dynamics of herbaceous by integrating additional environmental factors and advanced methodologies.

## Figures and Tables

**Figure 1 biology-14-01719-f001:**
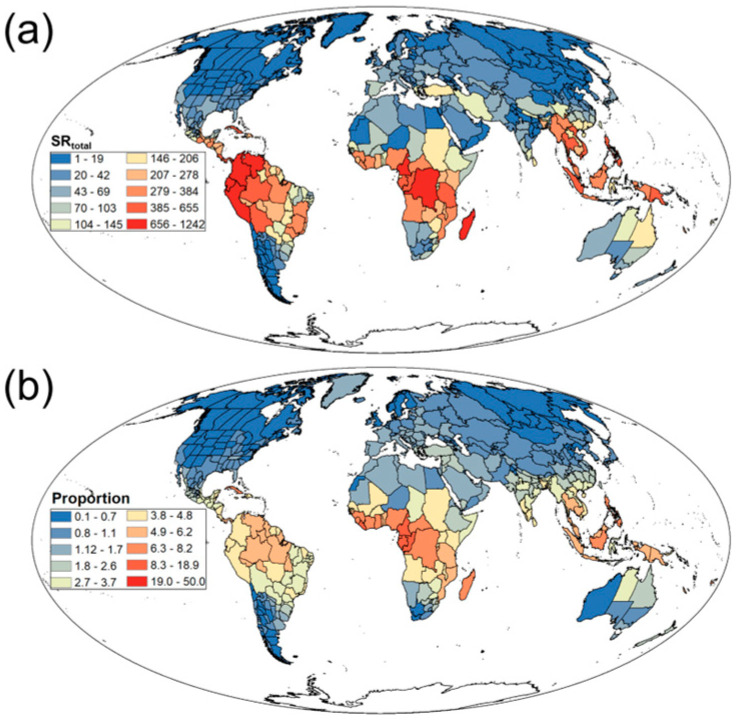
Geographical patterns of (**a**) total Rubiaceae species richness (SR_total_) and (**b**) the proportion of Rubiaceae species richness to total vascular plant species richness (Proportion).

**Figure 2 biology-14-01719-f002:**
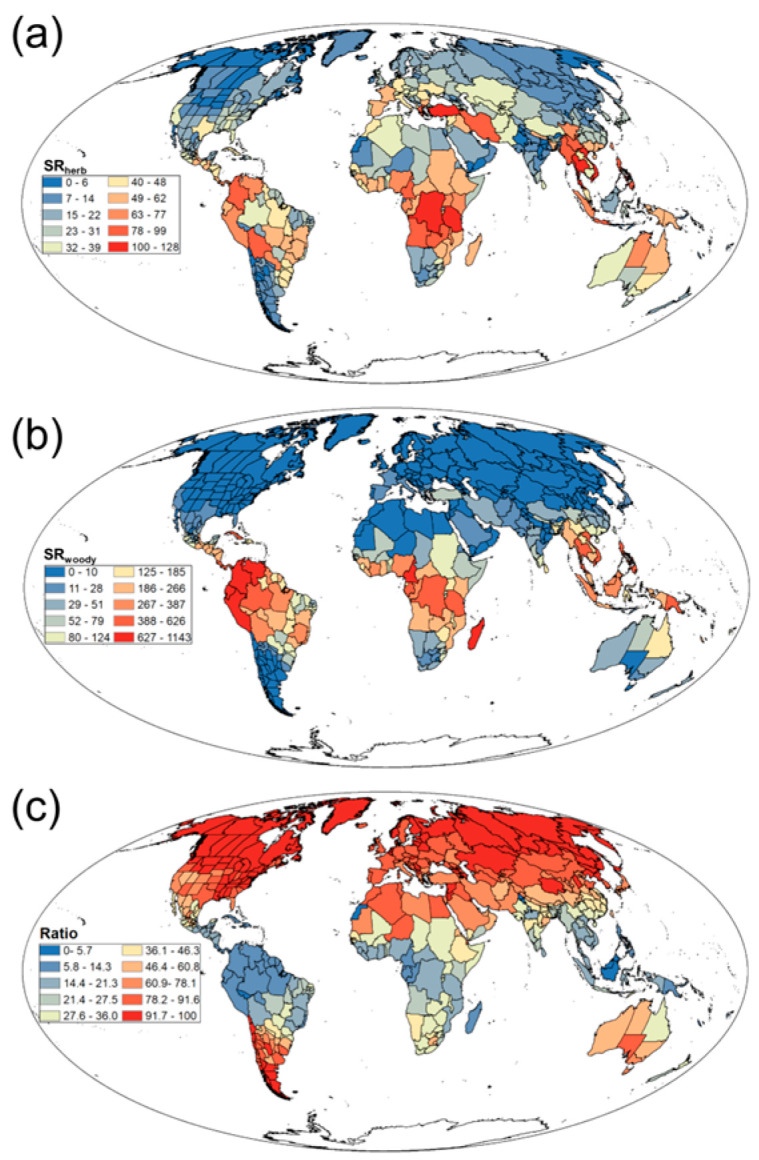
Geographical patterns of (**a**) herbaceous Rubiaceae species richness (SR_herb_), (**b**) woody Rubiaceae species richness (SR_woody_), and (**c**) the ratio of herbaceous Rubiaceae species richness to total Rubiaceae species richness (Ratio).

**Figure 3 biology-14-01719-f003:**
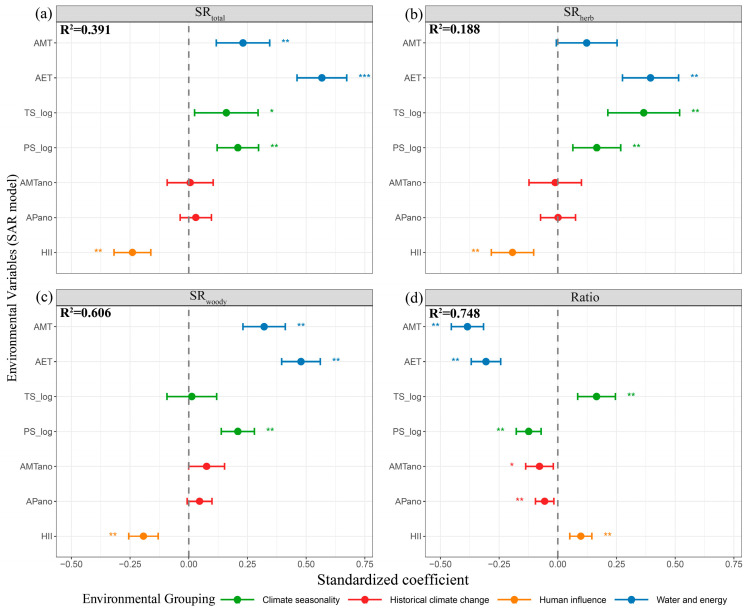
Standardized regression coefficients between different species richness components and seven environmental variables based on SAR analysis. (**a**) Total Rubiaceae species richness (SR_total_); (**b**) herbaceous Rubiaceae species richness (SR_herb_); (**c**) woody Rubiaceae species richness (SR_woody_); (**d**) the ratio of herbaceous Rubiaceae species richness to total Rubiaceae species richness (Ratio). The blue line represents water and energy, the green line represents climate seasonality, the red line represents historical climate change, and the yellow line represents human influence. The environmental factors on the *Y*-axis include Annual Mean Temperature (AMT), Actual Evapotranspiration (AET), Temperature Seasonality (log-transformed) (TS-log), Precipitation Seasonality (log-transformed) (PS-log), AMT anomaly (AMTano), AP anomaly (APano), and Human Influence Index (HII), with * indicating significant effects. ** and *** denote highly significant effects and extremely significant effects, respectively.

**Figure 4 biology-14-01719-f004:**
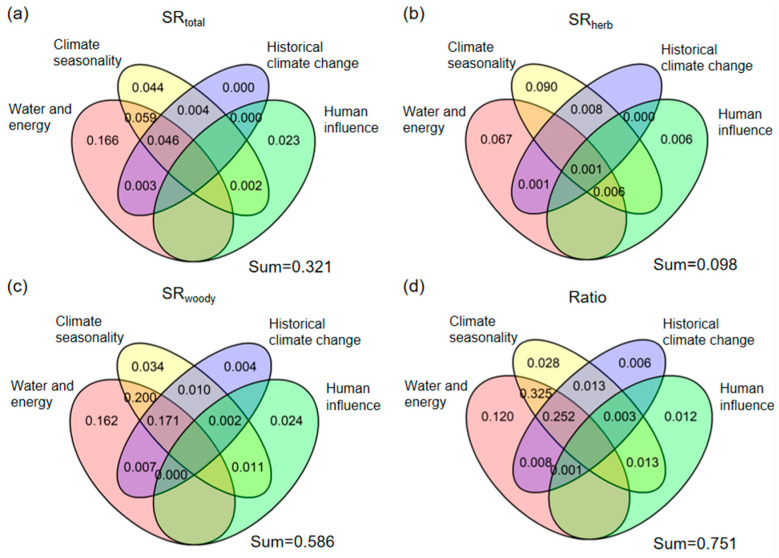
Partitioning the variation in (**a**) Total Rubiaceae species richness (SR_total_), (**b**) herbaceous Rubiaceae species richness (SR_herb_), (**c**) woody Rubiaceae species richness (SR_woody_) and (**d**) the ratio of herbaceous Rubiaceae species richness to total Rubiaceae species richness (Ratio) among environmental variables in four groups (water and energy, climate seasonality, historical climate change, and human influence variables).

## Data Availability

The original contributions presented in this study are included in the article/Appendix A. Further inquiries can be directed to the corresponding authors.

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
