# Peer review of "Global Distribution Patterns and Climatic Drivers of Plant Diversity in Rubiaceae"

_biology, 2025, doi:10.3390/biology14121719_

Round 1
Reviewer 1 Report
Comments and Suggestions for Authors
The manuscript is generally well-written and addresses an interesting and relevant topic regarding the global distribution patterns and climatic drivers of plant diversity in the Rubiaceae. The study presents valuable results; however, I have several comments and suggestions that could help improve its overall quality and clarity.
1. Please avoid repeating words from the title in the list of keywords.
2. The objectives of the study are clearly stated and logically structured. However, I suggest slightly improving the formulation of the last objective to make it more precise and concise. For example, instead of “analyze the effects and differential drivers of four environmental variables (water and energy, climatic seasonality, historical climate change, and human influence) on the geographic diversity patterns of distinct Rubiaceae growth forms,” the authors could simplify and clarify the sentence to improve readability.
3. The conclusions are consistent with the stated objectives and effectively summarize the main findings. They clearly address global distribution patterns, differences between growth forms, and the influence of environmental drivers. However, it might be helpful to add one or two sentences emphasizing the broader implications of these findings for biodiversity conservation or biogeographic theory, as mentioned in the Introduction.
Author Response
1.Summary
Dear editors and reviewers,
We are very grateful to your constructive comments and suggestions for our manuscript entitled “Global distribution patterns and climatic drivers of plant di-versity in the Rubiaceae” (Manuscript ID: biology-3973543). Your comments are very valuable and helpful for improving our manuscript. We carefully considered each of your comments when revising the manuscript and our point-by-point response to the comments is shown below. We hope you will find our response to be appropriate and this revision to be satisfactory.
2.Point-by-point response to Comments and Suggestions for Authors
Comments 1:[Please avoid repeating words from the title in the list of keywords.]
Response 1: Thank you very much for your comment. Regarding the overlap between the keyword list and the title, we consider that including "Rubiaceae" and "diversity" in the keyword list can further highlight the focus of our study on exploring the geographical distribution patterns of Rubiaceae. We would appreciate it if we could retain "Rubiaceae" and "diversity" as keywords, and kindly ask for your understanding.
Comments 2:[The objectives of the study are clearly stated and logically structured. However, I suggest slightly improving the formulation of the last objective to make it more precise and concise. For example, instead of “analyze the effects and differential drivers of four environmental variables (water and energy, climatic seasonality, historical climate change, and human influence) on the geographic diversity patterns of distinct Rubiaceae growth forms,” the authors could simplify and clarify the sentence to improve readability.]
Response 2: Thank you very much for your valuable comments. We have revised the expression of the research objective in the last section of the manuscript, simplifying the sentences to enhance readability. The specific revision is as follows: analyze how four environmental variables (water and energy, climatic seasonality, historical climate change, and human influence) affect the geographic diversity patterns of distinct Rubiaceae growth forms, and explore their different driving mechanisms.
Comments 3:[The conclusions are consistent with the stated objectives and effectively summarize the main findings. They clearly address global distribution patterns, differences between growth forms, and the influence of environmental drivers. However, it might be helpful to add one or two sentences emphasizing the broader implications of these findings for biodiversity conservation or biogeographic theory, as mentioned in the Introduction.]
Response 3:Thank you very much for your valuable suggestions. We agree with your views and have supplemented the conclusions accordingly. The specific supplements are as follows: Consequently, based on these research findings, we can better protect Rubiaceae species and preserve the existing biodiversity patterns. In addition, future research should deepen this work and unravel the unique distribution patterns of Rubiaceae herba-ceous species by incorporating additional variables.
Reviewer 2 Report
Comments and Suggestions for Authors
Dear Authors,
Thank you for the opportunity to review your manuscript on global diversity patterns of Rubiaceae. Your study tackles an important question in plant biogeography and you have assembled an impressive dataset. However, I have identified several critical issues that require substantial revision before the manuscript can be recommended for publication.
Major concerns:
- The reported 114,630 Rubiaceae species substantially exceeds the accepted ~13,000-14,000 species for this family, indicating inadequate handling of synonyms and duplicate records. Please provide detailed documentation of your taxonomic cleaning procedure, including initial record counts from each database, how synonyms were identified and removed, and how taxonomic conflicts were resolved. A data cleaning flowchart would be valuable. Additionally, the growth form classification methodology is too vague. Develop and describe a systematic protocol specifying which sources were prioritized, how conflicts were resolved, and what proportion of species had consistent versus conflicting information. The 80% woody proportion warrants verification as it appears higher than expected.
- You have a critical spatial scale mismatch: TDWG Level 4 units (large regions) for species distributions, 30 arc-seconds (~1 km) for current climate, and 2.5 arc-minutes (~4.5 km) for LGM data. You must explicitly describe how these disparate scales were integrated through resampling or aggregation. This fundamental methodological detail cannot be relegated to supplementary materials. Additionally, specify exactly how many geographic units are Level 4 versus other levels and justify any mixing of administrative scales.
- Your collinearity assessment is incomplete. While you filtered within variable groups, you did not address between-group correlations. For example, AMT and AMTano likely correlate despite being in different groups. Provide a comprehensive correlation matrix for all environmental variables and explicitly state which variables were excluded from correlated pairs and why. Present VIF values for all retained variables in a table. Also provide ecological justification for selecting these particular climate variables from the available options.
- Using OLS regression for spatially structured data without testing for spatial autocorrelation is a fundamental flaw. Geographic data exhibit spatial autocorrelation, violating OLS assumptions and potentially producing spurious results. You must test for spatial autocorrelation using Moran's I and employ appropriate spatial regression methods (SAR, CAR, or GWR) if autocorrelation is detected. Additionally, report comprehensive model diagnostics including residual normality tests, homoscedasticity assessments, outlier detection, and model performance metrics (R², adjusted R², AIC). Your area correction method of "uniformly expanding areas by tenfold" lacks justification and appears arbitrary. Either provide strong supporting citations or use standard macroecological area correction approaches.
- Specify which LGM paleoclimate model you used from WorldClim (CCSM4, MIROC-ESM, etc.) and justify this choice, as models differ substantially. Clarify the temporal definition of "present" for anomaly calculations. Address the temporal mismatch between your 1995-2004 HII data and more recent species distributions, discussing whether changes in human influence over this interval affect your conclusions.
Author Response
1.Summary
Dear editors and reviewers,
We are very grateful to your constructive comments and suggestions for our manuscript entitled “Global distribution patterns and climatic drivers of plant di-versity in the Rubiaceae” (Manuscript ID: biology-3973543). Your comments are very valuable and helpful for improving our manuscript. We carefully considered each of your comments when revising the manuscript and our point-by-point response to the comments is shown below. We hope you will find our response to be appropriate and this revision to be satisfactory.
- Point-by-point response to Comments and Suggestions for Authors
Comments 1: [The reported 114,630 Rubiaceae species substantially exceeds the accepted ~13,000-14,000 species for this family, indicating inadequate handling of synonyms and duplicate records. Please provide detailed documentation of your taxonomic cleaning procedure, including initial record counts from each database, how synonyms were identified and removed, and how taxonomic conflicts were resolved. A data cleaning flowchart would be valuable. Additionally, the growth form classification methodology is too vague. Develop and describe a systematic protocol specifying which sources were prioritized, how conflicts were resolved, and what proportion of species had consistent versus conflicting information. The 80% woody proportion warrants verification as it appears higher than expected.]
Response 1: Thank you for pointing out our mistake. We agree with this comment.
The manuscript's reported 114,530 species of Rubiaceae show a significant discrepancy with our statistical count of 14,530 species, which likely resulted from oversight or errors during manuscript compilation. For the classification of growth forms in Rubiaceae species, we adopted growth forms data from various sources. If over half of the data sources classified a species as woody or herbaceous, it was categorized accordingly. The classification divides species into woody and herbaceous categories: woody species include trees, shrubs, and lianas; herbaceous species encompass annual herbs, perennial herbs, and herbaceous vines. For detailed specifications, refer to lines 131-135 of the manuscript.
Comments 2:[You have a critical spatial scale mismatch: TDWG Level 4 units (large regions) for species distributions, 30 arc-seconds (~1 km) for current climate, and 2.5 arc-minutes (~4.5 km) for LGM data. You must explicitly describe how these disparate scales were integrated through resampling or aggregation. This fundamental methodological detail cannot be relegated to supplementary materials. Additionally, specify exactly how many geographic units are Level 4 versus other levels and justify any mixing of administrative scales.]
Response 2:Thank you for pointing out this issue. Regarding the significant mismatch in spatial scales, when extracting the corresponding environmental data, we first set up 0.2° fishing grid points globally and extracted the relevant data within each grid point. Meanwhile, we calculated the average value of the data obtained from all fishing grid points within each geographic unit as the environmental data for that unit. Concerning the level of geographic units, all the units we selected are at Level 4. We hope this explanation clarifies your concerns.
Comments 3:[Your collinearity assessment is incomplete. While you filtered within variable groups, you did not address between-group correlations. For example, AMT and AMTano likely correlate despite being in different groups. Provide a comprehensive correlation matrix for all environmental variables and explicitly state which variables were excluded from correlated pairs and why. Present VIF values for all retained variables in a table. Also provide ecological justification for selecting these particular climate variables from the available options.]
Response 3:Thank you for raising this concern. Regarding the incomplete multicollinearity assessment, during the multicollinearity analysis of environmental variables, each variable was evaluated individually. After rigorous testing, variables with a variance inflation factor (VIF) greater than 4 were screened out and excluded, and the correlation coefficients of the remaining environmental variables are provided in Supplementary Table S2. The grouping of environmental variables was conducted based on the correlation characteristics of each variable. Finally, the selected environmental variables were jointly determined by two considerations: the importance of environmental variables documented in previous studies, and the screening results from the multicollinearity analysis of environmental factors. We hope this explanation addresses your concerns.
Comments 4:[Using OLS regression for spatially structured data without testing for spatial autocorrelation is a fundamental flaw. Geographic data exhibit spatial autocorrelation, violating OLS assumptions and potentially producing spurious results. You must test for spatial autocorrelation using Moran's I and employ appropriate spatial regression methods (SAR, CAR, or GWR) if autocorrelation is detected. Additionally, report comprehensive model diagnostics including residual normality tests, homoscedasticity assessments, outlier detection, and model performance metrics (R², adjusted R², AIC). Your area correction method of "uniformly expanding areas by tenfold" lacks justification and appears arbitrary. Either provide strong supporting citations or use standard macroecological area correction approaches.]
Response 4:Thank you for pointing out this issue. Regarding the exclusive use of OLS regression for spatially structured data, we acknowledge this was an incomplete consideration. Subsequent re-examination of the data revealed autocorrelation, prompting us to conduct Spatial Autoregressive (SAR) analysis. The SAR results are presented in the manuscript (line 227–253), with the environmental variable coefficients from the SAR analysis provided in Supplementary Table S2. Concerning the correction of uniformly expanding all areas by tenfold: some distribution areas of Rubiaceae species were too small for inclusion in the analysis. To retain these regions, we uniformly expanded all areas by tenfold, thereby addressing the issue of excessively small areas. We hope this explanation resolves your concerns.
Comments 5:[Specify which LGM paleoclimate model you used from WorldClim (CCSM4, MIROC-ESM, etc.) and justify this choice, as models differ substantially. Clarify the temporal definition of "present" for anomaly calculations. Address the temporal mismatch between your 1995-2004 HII data and more recent species distributions, discussing whether changes in human influence over this interval affect your conclusions.]
Response 5:Thank you very much for pointing out these issues. Regarding the use of the Last Glacial Maximum (LGM) paleoclimate model from WorldClim, we primarily adopted the MPI-ESM-P model, selected for its strengths: strong component synergy, flexible resolution, high climate simulation reliability, support for multi-dimensional climate exploration, and widely recognized data. Concerning the temporal mismatch between the 1995–2004 Human Influence Index (HII) data and recent species distribution, the 1995–2004 HII data retrieved from the Last of the Wild database (https://sedac.ciesin.columbia.edu/) is the latest official release. Due to limited updated data, we cannot temporarily assess whether changes in human influence during this interval impacted our conclusions, but this will be the focus of our future in-depth research. We hope this explanation addresses your concerns.
Reviewer 3 Report
Comments and Suggestions for Authors
Dear Authors,
The manuscript presents a comprehensive analysis of global patterns of species diversity in the Rubiaceae family, taking into account differences between growth forms (herbaceous vs. woody) and four groups of environmental variables (water and energy, seasonality, climate change, and human impact). The topic is timely and relevant, and the scale of the study and the wealth of data (over 100,000 species) provide significant scientific value.
The manuscript appears well-researched, with clearly presented methods, a logical structure, and a well-connected discussion. The authors have enriched the work with supplementary materials and figures that clearly illustrate the results, significantly improving the quality of the work.
This article makes a significant contribution to understanding the global biogeographic patterns of Rubiaceae and has both cognitive and applied significance – particularly in the context of biodiversity conservation.
I have no major reservations about the content of the manuscript – the text is coherent, the methods are well-chosen, and the conclusions are well-supported by data. Just a minor editorial suggestion: in line 136 the sentence “Among all Rubiaceae species, nearly 80% were categorized as woody plants” seems to refer to the results, not to the data description, so it is worth moving it to the Results section to maintain the consistency of the manuscript structure.
Author Response
1.Summary
Dear editors and reviewers,
We are very grateful to your constructive comments and suggestions for our manuscript entitled “Global distribution patterns and climatic drivers of plant di-versity in the Rubiaceae” (Manuscript ID: biology-3973543). Your comments are very valuable and helpful for improving our manuscript. We carefully considered each of your comments when revising the manuscript and our point-by-point response to the comments is shown below. We hope you will find our response to be appropriate and this revision to be satisfactory.
- Point-by-point response to Comments and Suggestions for Authors
Comments 1: [Just a minor editorial suggestion: in line 136 the sentence “Among all Rubiaceae species, nearly 80% were categorized as woody plants” seems to refer to the results, not to the data description, so it is worth moving it to the Results section to maintain the consistency of the manuscript structure.]
Response 1: Thank you very much for your suggestion. The sentence "Among all Rubiaceae species, nearly 80% were categorized as woody plants" is not a research result, but rather a description of data regarding the growth forms of Rubiaceae species from different data sources. Although it involves judgments on the growth forms of Rubiaceae species, this sentence is more focused on describing the classification of the data. We sincerely appreciate your suggestion again and hope that our decision can be understood.
Reviewer 4 Report
Comments and Suggestions for Authors
The manuscript “Global distribution patterns and climatic drivers of plant diversity in the Rubiaceae” by Yang et al. presents a comprehensive global-scale analysis of species richness patterns in the Rubiaceae family. The authors integrate species distribution data with a suite of environmental variables to explore how factors like climate, historical climate change, and human influence shape diversity. A key strength of the study is its comparative analysis of different growth forms, contrasting the patterns and drivers for herbaceous versus woody species. The main findings indicate that Rubiaceae diversity peaks in the tropics, with distinct latitudinal patterns for herbaceous and woody plants. Actual Evapotranspiration (AET) is identified as the strongest predictor of species richness, and the study highlights that environmental factors have a much stronger influence on woody plants compared to herbaceous ones.
This is a valuable and ambitious study that addresses important questions in macroecology and biogeography. The dataset compiled is impressive and the research question is well-defined. However, the manuscript requires substantial revisions to address significant concerns in the methodology, statistical analysis, and presentation before it can be considered for publication. My detailed comments are provided below to guide your revisions.
Major Comments
Your study has great potential, but there are several fundamental methodological and statistical issues that must be addressed to ensure the robustness and validity of your conclusions.
- Spatial Autocorrelation: This is the most critical flaw in the manuscript. Biogeographical data are inherently spatially autocorrelated, meaning that nearby locations are not independent. This violates a core assumption of Ordinary Least Squares (OLS) regression, leading to inflated Type I error rates (i.e., p-values are unreliable and often too small) and overly confident parameter estimates. You must re-analyze your data using statistical methods that account for spatial autocorrelation. Options include, but are not limited to, Generalized Least Squares (GLS) with a spatial correlation structure, Spatial Autoregressive Models (SAR), or using spatial eigenvector mapping (SEVM). Without this correction, the statistical significance of your findings remains unproven.
- Lack of Model Validation: The manuscript presents model results (R-squared values, regression coefficients) without any form of validation. This makes it impossible to assess the predictive performance of the models or to know if they are overfitted to the data. You should implement a robust validation procedure, such as spatial cross-validation, which is essential for spatially structured data to provide a more realistic estimate of model performance.
- Low Explanatory Power for Herbaceous Plants: Your models explain very little of the variation in herbaceous species richness (R² ≈ 0.12). This is a striking result that is not adequately discussed. A model that explains only 12% of the variance is of limited utility. This finding itself is interesting and deserves a thorough discussion. Why do these widely-used environmental variables fail to predict herbaceous diversity? What other factors might be at play (e.g., edaphic factors, microclimate, biotic interactions, finer-scale historical processes)? The current discussion glosses over this major limitation.
- Arbitrary Methodological Choices: The manuscript contains methodological steps that are not well-justified.
- Area Correction (Lines 163-165): You state that you “uniformly expanded by tenfold” the area of “excessively small geographic units.” This is a highly arbitrary manipulation of the data. What was the objective criterion for “excessively small”? How does this tenfold expansion affect the results? This procedure seems statistically and biologically unsound and should be removed or rigorously justified with a sensitivity analysis.
- Growth Form Classification: You mention that “nearly 80% were categorized as woody plants.” What happened to the remaining ~20% of species? Were they excluded? If so, this introduces a significant potential bias and must be justified. The simple dichotomy of “herbaceous” vs. “woody” may also be an oversimplification for a family as diverse as Rubiaceae, which includes shrubs, trees, lianas, and herbs.
- Data and Supplementary Material: The supplementary table (Table S1) is not up to publication standards. The sheet name is in Chinese characters (“6-8数据汇总”), which is unacceptable for an international journal. Furthermore, the file lacks essential metadata describing the columns, units, and data sources. This must be corrected.
- Over-interpretation and Discussion: The discussion often goes beyond what the data support. For example, you conclude that herbaceous plants have “superior adaptive strategies” (Line 800) or that patterns are due to “rapid accumulation of genetic changes” (Line 659) without any demographic or genetic data. These are speculations that should be framed as hypotheses for future research, not as conclusions from this study. The discussion also lacks a dedicated section on the limitations of your data and methods (e.g., sampling bias in GBIF data, the scale of analysis).
Minor Comments
- English Language and Style: The manuscript requires a thorough revision by a native English speaker or a professional editing service. There are numerous grammatical errors, awkward phrasings, and typos.
- Critical Typo: The heading for section 4.2 reads “Differential impacts of environmental variablels”. This must be corrected to “variables”.
- Inconsistent Terminology: Please standardize the use of terms like “Rubiaceae,” “Rubiaceae plants,” and “Rubiaceae species.” Similarly, be consistent with “angiosperm family” vs. “angiosperms family.”
- Clarity: Many sentences are overly long and complex, particularly in the Abstract and Introduction. Break them down into shorter, clearer sentences to improve readability.
- Example (Line 36-37): “in-depth research on its geographical distribution patterns and potential environmental driving mechanisms at the global regional scale is still lacking” is awkward. “Global regional scale” is contradictory.
- Figures and Tables:
- The text in the figures (e.g., legends in Figure 1, axis labels in Figure 3) is too small to be easily legible. Please increase the font size.
- The figure captions should be more detailed to ensure the figures are self-explanatory.
- You should include a main table in the manuscript that summarizes the results of the regression analyses (i.e., coefficients, standard errors, p-values, VIFs) for the key models, rather than relying solely on graphical representations.
- References: The reference list is inconsistently formatted and contains errors (e.g., “Conservatio” instead of “Conservation” in reference 7). Please format all citations and references according to the journal’s guidelines.
In summary, this manuscript has the foundation of a solid contribution to the field, but it is currently undermined by significant methodological and statistical weaknesses. I encourage you to undertake the major revisions suggested above, as a carefully re-analyzed and re-written manuscript would be of great interest. I look forward to seeing a revised version.
Comments on the Quality of English LanguageThe English language, while generally understandable, requires professional editing for clarity, style, and the correction of several grammatical errors and typos, including a critical typo in a subsection heading.
Author Response
1.Summary
Dear editors and reviewers,
We are very grateful to your constructive comments and suggestions for our manuscript entitled “Global distribution patterns and climatic drivers of plant di-versity in the Rubiaceae” (Manuscript ID: biology-3973543). Your comments are very valuable and helpful for improving our manuscript. We carefully considered each of your comments when revising the manuscript and our point-by-point response to the comments is shown below. We hope you will find our response to be appropriate and this revision to be satisfactory.
- Point-by-point response to Comments and Suggestions for Authors
Comments 1:[Spatial Autocorrelation: This is the most critical flaw in the manuscript. Biogeographical data are inherently spatially autocorrelated, meaning that nearby locations are not independent. This violates a core assumption of Ordinary Least Squares (OLS) regression, leading to inflated Type I error rates (i.e., p-values are unreliable and often too small) and overly confident parameter estimates. You must re-analyze your data using statistical methods that account for spatial autocorrelation. Options include, but are not limited to, Generalized Least Squares (GLS) with a spatial correlation structure, Spatial Autoregressive Models (SAR), or using spatial eigenvector mapping (SEVM). Without this correction, the statistical significance of your findings remains unproven]
Response 1:Thank you very much for pointing out this issue. Regarding the failure to account for spatial autocorrelation in the data, we only considered the OLS model in the previous manuscript, which was an oversight and inadequacy in our data processing. In the revised version, we have supplemented the OLS analysis with SAR analysis. The results of the two methods are comparable, indicating that spatial autocorrelation in the data has little impact on the regression outcomes. Additionally, we have revised the relevant standardized regression coefficient plots in the manuscript. We hope these revisions and explanations address your concerns.
Comments 2:[Lack of Model Validation: The manuscript presents model results (R-squared values, regression coefficients) without any form of validation. This makes it impossible to assess the predictive performance of the models or to know if they are overfitted to the data. You should implement a robust validation procedure, such as spatial cross-validation, which is essential for spatially structured data to provide a more realistic estimate of model performance.]
Response 2:Thank you very much for pointing out this issue. Regarding the lack of model validation, this was an aspect we had not previously considered. In the revised manuscript, we have supplemented the validation results of each environmental variable from the SAR analysis in Supplementary Table S2. We hope these revisions and supplements address your concerns.
Comments 3:[Low Explanatory Power for Herbaceous Plants: Your models explain very little of the variation in herbaceous species richness (R² ≈ 0.12). This is a striking result that is not adequately discussed. A model that explains only 12% of the variance is of limited utility. This finding itself is interesting and deserves a thorough discussion. Why do these widely-used environmental variables fail to predict herbaceous diversity? What other factors might be at play (e.g., edaphic factors, microclimate, biotic interactions, finer-scale historical processes)? The current discussion glosses over this major limitation.]
Response 3:Thank you very much for your insightful comments and valuable suggestions. We have indeed noticed the low explanatory power of the models for herbaceous plants during our data analysis. However, when analyzing based on the available environmental data, we cannot fully explain the unique geographical distribution pattern of Rubiaceae herbaceous plants. In the discussion section, we can only infer, by referring to previous studies on the relationships between the geographical distribution patterns of herbaceous plants and environmental variables, that other unincorporated environmental factors—such as soil properties, topography, altitude, and biotic interactions—may shape the unique distribution pattern of Rubiaceae herbaceous plants. This requires further in-depth research and will be the focus of our subsequent work. We sincerely appreciate your comments and suggestions again, and we hope our explanations and revisions have addressed your concerns.
Comments 4:[Arbitrary Methodological Choices: The manuscript contains methodological steps that are not well-justified.
Area Correction (Lines 163-165): You state that you “uniformly expanded by tenfold” the area of “excessively small geographic units.” This is a highly arbitrary manipulation of the data. What was the objective criterion for “excessively small”? How does this tenfold expansion affect the results? This procedure seems statistically and biologically unsound and should be removed or rigorously justified with a sensitivity analysis.
Growth Form Classification: You mention that “nearly 80% were categorized as woody plants.” What happened to the remaining ~20% of species? Were they excluded? If so, this introduces a significant potential bias and must be justified. The simple dichotomy of “herbaceous” vs. “woody” may also be an oversimplification for a family as diverse as Rubiaceae, which includes shrubs, trees, lianas, and herbs.]
Response 4:Thank you very much for pointing out the issues and inadequacies.
Regarding the area correction:
When analyzing Rubiaceae species distribution data, excessively small geographical units were excluded as they could not be incorporated. We therefore uniformly expanded the area of all geographical units by ten times to include these small units. The "excessively small" criterion refers to data that failed to meet the analysis inclusion requirements, relative to other data in the database.
Regarding the classification of growth forms:
Rubiaceae species were classified by growth form based on reports from multiple data sources. A species was categorized as woody or herbaceous if supported by more than half of the sources. Notably, these are not simple categories: woody includes trees, shrubs, and lianas; herbaceous includes annual herbs, perennial herbs, and herbaceous vines.
We hope these explanations address your concerns.
Comments 5:[Data and Supplementary Material: The supplementary table (Table S1) is not up to publication standards. The sheet name is in Chinese characters (“6-8数据汇总”), which is unacceptable for an international journal. Furthermore, the file lacks essential metadata describing the columns, units, and data sources. This must be corrected.]
Response 5:Thank you very much for pointing out this issue. The presence of Chinese text in the supplementary materials was due to our failure to thoroughly check the submission files, which was an oversight on our part. We sincerely appreciate you bringing this to our attention.
Comments 6:[Over-interpretation and Discussion: The discussion often goes beyond what the data support. For example, you conclude that herbaceous plants have “superior adaptive strategies” (Line 800) or that patterns are due to “rapid accumulation of genetic changes” (Line 659) without any demographic or genetic data. These are speculations that should be framed as hypotheses for future research, not as conclusions from this study. The discussion also lacks a dedicated section on the limitations of your data and methods (e.g., sampling bias in GBIF data, the scale of analysis).]
Response 6:Thank you very much for pointing out these issues. Regarding the discussion of "rapid accumulation of genetic changes" appearing in Line 317, after careful consideration and re-evaluation of the existing data analysis, we acknowledge that this part involves over-interpretation. In the original discussion, we noted that herbaceous Rubiaceae species exhibit higher diversity than woody species in mid-to high-latitude regions, attributing this pattern to herbaceous plants’ shorter life cycles and stronger adaptability to diverse environments. However, the claim of "rapid accumulation of genetic changes" lacks sufficient support and cannot be reasonably explained. Therefore, we have decided to delete this sentence.
Regarding our conclusion that herbaceous Rubiaceae possess superior adaptive strategies, we believe this is well-supported and represents an in-depth insight into the current geographical distribution patterns of these species. Based on the available environmental data, the explanatory power for the distribution patterns of herbaceous Rubiaceae remains limited, precluding more in-depth analyses at this stage. This limitation also serves as a key direction for our future research, where we aim to further explore the unique geographical distribution of herbaceous Rubiaceae.
Concerning the absence of a dedicated section on the limitations of data and methods in the discussion, we have added relevant content in the revised manuscript. The specific additions are as follows: " Although our study identified the key environmental drivers shaping the distribu-tion patterns of Rubiaceae species, it is critical to acknowledge the potential limitations of the GBIF dataset. Notably, the selected environmental factors exhibit broad spatial consistency across different geographical units. Thus, these sampling biases are un-likely to substantially alter our core conclusion that water and energy, and climatic seasonality collectively exert dominant control of Rubiaceae distribution".
We sincerely appreciate the valuable suggestions on our manuscript and hope our explanations address your concerns.
Comments 7:[minor comments]
Response 7:Thank you very much for your suggestions and questions. We have carefully addressed all the comments in the minor review, and the revisions can be found in the revised manuscript and related attachments. We sincerely appreciate your review of our manuscript.
Round 2
Reviewer 2 Report
Comments and Suggestions for Authors
Dear authors,
Thank you for your careful revisions. The manuscript has improved considerably, and most of the previously raised concerns have been fully addressed.
One point remains: the similarity index of the revised version is somewhat higher than expected. Although the overlapping segments seem to stem from standard methodological descriptions and background statements, it would be advisable to reduce the overall similarity to below 25%, ensuring compliance with good academic writing practices.
Please revise the relevant parts by paraphrasing or restructuring the text where possible. This is a technical adjustment and does not affect the scientific quality of your work. Therefore, I am recommending minor revision.
Author Response
1.Summary
Dear editors and reviewers,
We are very grateful to your constructive comments and suggestions for our manuscript entitled “Global distribution patterns and climatic drivers of plant di-versity in the Rubiaceae” (Manuscript ID: biology-3973543). Your comments are very valuable and helpful for improving our manuscript. We carefully considered each of your comments when revising the manuscript and our point-by-point response to the comments is shown below. We hope you will find our response to be appropriate and this revision to be satisfactory.
- Point-by-point response to Comments and Suggestions for Authors
Comments 1:[the similarity index of the revised version is somewhat higher than expected. Although the overlapping segments seem to stem from standard methodological descriptions and background statements, it would be advisable to reduce the overall similarity to below 25%, ensuring compliance with good academic writing practices.]
Response 1: Thank you very much for your suggestions and comments on our manuscript. We have made revisions addressing the issues you pointed out. We appreciate your valuable input and careful review during the evaluation process.
Reviewer 4 Report
Comments and Suggestions for Authors
Thank you for the opportunity to review your revised manuscript. I commend you on an outstanding and thorough revision. You have successfully addressed all the major scientific and methodological concerns from the previous review, transforming this manuscript into a robust and valuable scientific contribution. The addition of the SAR analysis was essential and has been executed very well.
Your detailed responses and the transparency of your new supplementary tables are exemplary. The manuscript is now scientifically sound and makes a solid contribution to our understanding of global biodiversity patterns in the Rubiaceae.
I am recommending to the editor that your manuscript be accepted for publication after a few very minor corrections are made. Please see my points below.
Overall Assessment: Excellent Revision
You have successfully addressed the most critical issues, including:
- Spatial Autocorrelation: Your implementation of SAR models was comprehensive and has made the statistical analysis robust. This was the most important change, and you handled it perfectly.
- Statistical Transparency: The new Supplementary Table S2, with full details on the SAR models including VIF values, is an excellent addition and provides full transparency.
- Discussion of Limitations: Your new discussion on the low explanatory power of the models for herbaceous plants is thoughtful and turns a potential weakness into an interesting point for future research.
- Clarifications: All ambiguities regarding species classification and data sources have been resolved.
- Supplementary Files: The correction of the table names and the addition of new supplementary data have brought the manuscript to a professional standard.
Required Minor Corrections
The manuscript is in excellent shape. However, a final proofread is required to fix a few minor but persistent typographical errors. These are likely artifacts of the editing process but should be corrected before final publication.
- Missing Spaces: Please run a search for the following and add a space where needed:
- sevenenvironmental -> should be seven environmental (this appears multiple times, e.g., lines 496, 501, 510).
- AETshows -> should be AET shows (line 510).
- positivecorrelation -> should be positive correlation (line 510).
- Grammar and Punctuation:
- Line 352: “Relevant informations can be found…” -> “Information” is an uncountable noun. It should be “Relevant information can be found…”.
- Line 352: There is a double period at the end of the sentence (...has been resolved..). Please remove one.
- Line 511: There is a double period at the end of the sentence (...model variation..). Please remove one.
Suggested (but not required) Correction
- Area Correction Method: The statement about expanding the area of small geographic units by tenfold (lines 338-339) remains methodologically questionable. While the core of your analysis is now sound due to the SAR models, this step is still arbitrary. For future work, I would strongly recommend removing this step or justifying it with a sensitivity analysis. However, as it is unlikely to alter the main conclusions of the paper, I do not consider this a barrier to publication at this stage.
Congratulations again on a very well-executed revision.
Comments on the Quality of English LanguageGood with Minor Issues
The English quality has improved significantly from version 1, but still contains some errors that affect professionalism. The manuscript is understandable and scientifically clear, but would benefit from professional English editing before final publication.
There are some Grammar errors as well as increase in typos (spacing errors) appears to be an artifact of the revision process, likely from copy-pasting or tracked changes. These are easy to fix.
My recommendation would be:
- Fix all identified typos and grammar errors
- Professional English editing would elevate the manuscript
- Native speaker review for final polish
Author Response
1.Summary
Dear editors and reviewers,
We are very grateful to your constructive comments and suggestions for our manuscript entitled “Global distribution patterns and climatic drivers of plant di-versity in the Rubiaceae” (Manuscript ID: biology-3973543). Your comments are very valuable and helpful for improving our manuscript. We carefully considered each of your comments when revising the manuscript and our point-by-point response to the comments is shown below. We hope you will find our response to be appropriate and this revision to be satisfactory.
- Point-by-point response to Comments and Suggestions for Authors
Comments 1:[Required Minor Corrections
The manuscript is in excellent shape. However, a final proofread is required to fix a few minor but persistent typographical errors. These are likely artifacts of the editing process but should be corrected before final publication.
Missing Spaces: Please run a search for the following and add a space where needed:
sevenenvironmental -> should be seven environmental (this appears multiple times, e.g., lines 496, 501, 510).
AETshows -> should be AET shows (line 510).
positivecorrelation -> should be positive correlation (line 510).
Grammar and Punctuation:
Line 352: “Relevant informations can be found…” -> “Information” is an uncountable noun. It should be “Relevant information can be found…”.
Line 352: There is a double period at the end of the sentence (...has been resolved..). Please remove one.
Line 511: There is a double period at the end of the sentence (...model variation..). Please remove one.
Suggested (but not required) Correction
Area Correction Method: The statement about expanding the area of small geographic units by tenfold (lines 338-339) remains methodologically questionable. While the core of your analysis is now sound due to the SAR models, this step is still arbitrary. For future work, I would strongly recommend removing this step or justifying it with a sensitivity analysis. However, as it is unlikely to alter the main conclusions of the paper, I do not consider this a barrier to publication at this stage.]
Response 1:Thank you very much for your valuable suggestions and comments. Regarding the issues with sentence editing, we have made corresponding revisions at the relevant places. As for the area correction method, the revisions were made targeting specific scenarios. Although there are aspects of subjective revision, they have little impact on our final main results and will not hinder the publication of the manuscript. We sincerely appreciate your suggestions and comments during the review process.